# Association of SARS-CoV-2 BA.4/BA.5 Omicron lineages with immune escape and clinical outcome

Joseph A. Lewnard [1,2,3] ✉, Vennis Hong[4], Jeniffer S. Kim[4], Sally F. Shaw [4], Bruno Lewin[4], Harpreet Takhar[4] & Sara Y. Tartof [4,5] ✉

Expansion of the SARS-CoV-2 BA.4 and BA.5 Omicron subvariants in populations with prevalent immunity from prior infection and vaccination, and associated burden of severe COVID-19, has raised concerns about epidemiologic characteristics of these lineages including their association with immune escape or severe clinical outcomes. Here we show that BA.4/BA.5 cases in a large US healthcare system had at least 55% (95% confidence interval: 43–69%) higher adjusted odds of prior documented infection than time-matched BA.2 cases, as well as 15% (9–21%) and 38% (27–49%) higher adjusted odds of having received 3 and ≥4 COVID-19 vaccine doses, respectively. However, after adjusting for differences in epidemiologic characteristics among cases with each lineage, BA.4/BA.5 infection was not associated with differential risk of emergency department presentation, hospital admission, or intensive care unit admission following an initial outpatient diagnosis. This finding held in sensitivity analyses correcting for potential exposure misclassification resulting from unascertained prior infections. Our results demonstrate that the reduced severity associated with prior (BA.1 and BA.2) Omicron lineages, relative to the Delta variant, has persisted with BA.4/BA.5, despite the association of BA.4/BA.5 with increased risk of breakthrough infection among previously vaccinated or infected individuals.

The SARS-CoV-2 Omicron (B.1.1.529) variant emerged in late 2021 and rapidly achieved global dissemination, accounting for a majority of incident SARS-CoV-2 infections within the United States by late December, 2021[1,2]. By February, 2022, 58% of US adults and 75% of US children aged ≤17 years were estimated to have acquired SARS-CoV-2 infection, with nearly half of these infections occurring during the initial expansion of the BA.1 subvariant lineage[3]. COVID-19 vaccination and naturally-acquired immunity from infection with pre-Omicron variants have generally been found to confer robust protection against

clinically severe disease involving the BA.1 lineage, although at weaker levels when compared with protection against pre-Omicron variants[4–7]. Thus, whereas expansion of the Omicron variant was associated with surges in COVID-19 hospitalizations and deaths, the proportion of Omicron cases resulting in severe illness has been lower than that experienced with prior variants and during periods with lower population-level immunity[4,8].

Following the initial peak in BA.1 infections within the US from December, 2021 to February, 2022, multiple Omicron lineages have

[1]Division of Epidemiology, School of Public Health, University of California, Berkeley, Berkeley, CA 94720, USA. [2]Division of Infectious Diseases & Vaccinology, School of Public Health, University of California, Berkeley, Berkeley, CA 94720, USA. [3]Center for Computational Biology, College of Engineering, University of California, Berkeley, Berkeley, CA 94720, USA. [4]Department of Research & Evaluation, Kaiser Permanente Southern California, Pasadena, CA 91101, USA. [5]Department of Health Systems Science, Kaiser Permanente Bernard J. Tyson School of Medicine, Pasadena, CA 91101, USA. ✉e-mail: jLewnard@berkeley.edu; Sara.Y.Tartof@kp.org

driven subsequent surges in cases, leading to persisting clinical burden and extended timetables for implementation of COVID-19 mitigation measures. Although not associated with enhanced severity or risk of breakthrough infection after vaccination or natural infection[4,9], the BA.2 lineage surpassed BA.1 in incident cases within the US beginning in March, 2022. Subsequently, the BA.4 and BA.5 lineages have become dominant globally[10]. BA.4 and BA.5 share a spike (S) protein harboring numerous mutations relative to BA.2, which may compromise the effectiveness of immune responses induced by prior infection and vaccination[11]. Other mutations specific to BA.4 and BA.5 alter binding to human angiotensin-converting enzyme-2 and non-S antibodies derived from prior infection[12,13]. However, clinical implications of the emergence of BA.4/BA.5 remain uncertain, as the burden of hospitalized and fatal COVID-19 cases observed during BA.4/BA.5 waves has varied widely across settings[14].

Monitoring the relative severity of infections caused by successive SARS-CoV-2 lineages, and their capacity to evade vaccine- or infection-derived immunity, is of key importance to informing public health mitigation measures as SARS-CoV-2 establishes endemic circulation. We, therefore, compared clinical outcomes and characteristics of contemporaneous cases with BA.2 and BA.4/BA.5 lineage Omicron variant infections within the Kaiser Permanente Southern California (KPSC) healthcare system from 29 April to 29 July, 2022[4]. As a comprehensive, integrated care organization, KPSC delivers healthcare across telehealth, outpatient, emergency department, and inpatient settings for over 4.7 million members. Electronic health records (EHRs) across all clinical settings, together with laboratory, pharmacy, and immunization data, record all care delivered by KPSC. These observations are augmented by insurance claims for out-of-network diagnoses, prescriptions, and procedures, enabling near-complete capture of healthcare interactions for KPSC members.

## Results

To compare severity of disease caused by BA.4/BA.5 and BA.2 infections, we monitored the frequency of healthcare interactions indicative of COVID-19 clinical progression occurring after an initial positive molecular SARS-CoV-2 test in any outpatient setting. Endpoints of interest included subsequent (≥1 day after testing) emergency department (ED) presentation or inpatient admission due to any cause, inpatient admission associated with an acute respiratory infection (ARI) diagnosis (Table S1), intensive care unit (ICU) admission, mechanical ventilation, and mortality. As KPSC implemented a home-based monitoring program for COVID-19 cases throughout the study period, with standardized criteria for ED referral and inpatient admission aiming to preserve hospital capacity, these endpoints provided consistent markers of disease progression within the sample followed from an initial outpatient test[15]. We restricted our analytic sample to individuals who first tested positive in an outpatient setting to select on healthcare-seeking behavior within the study population, thus maximizing internal validity when comparing outcomes among BA.4/BA.5 and BA.2 cases.

In total, 106,532 SARS-CoV-2 cases out of 148,105 diagnosed as outpatients at KPSC during the study period met eligibility criteria and were included in analyses. We excluded 18,799 patients without ≥1 year of continuous enrollment before their positive test, and 22,774 whose tests were not processed using the ThermoFisher TaqPath COVID-19 Combo Kit, which enabled lineage determination based on S gene target failure (SGTF; see Methods). Within a validation sample of specimens for which whole-genome sequencing results were available, 1595 of 1620 (98.5%) specimens with the S gene detected, and 1196 of 1252 (95.5%) specimens with SGTF, were confirmed to belong to the BA.2 and BA.4/BA.5 lineages, respectively (Table S2). Additionally, the likelihood of sequencing failure did not differ appreciably between SGTF samples and non-SGTF samples submitted for sequencing (1085/ 2335 [46.4%], SGTF vs. 1211/2830 [42.8%], non-SGTF), confirming that

SGTF was not an artifact of spurious dropout of the S gene probe in specimens with low viral RNA quantities (Table S3).

Within our analytic sample, 59,556 (55.9%) and 46,976 (44.1%) cases were considered infected with the BA.4/BA.5 and BA.2 lineages, respectively, based on SGTF. Whereas the weekly proportion of cases with BA.4/BA.5 infections expanded from 3.0% to 92.9% over the study period, the proportion progressing to hospital admission was stable over this period within the range of 0.2–0.5%, as compared to 0.7–1.0% during February, 2022 (Fig. 1). Age distributions were similar among cases infected with either lineage, with 13.1% of all cases aged 0–17 years, 31.9% aged 18–39 years, 42.6% aged 40–64 years, and 12.4% aged ≥65 years (Table 1). Other case attributes, including race/ethnicity, sex, body mass index, Charlson comorbidity index, neighborhood socioeconomic status, prior-year healthcare utilization, and receipt of vaccines targeting respiratory pathogens other than SARS-CoV-2 did not differ markedly between cases infected with the BA.4/BA.5 and BA.2 lineages, facilitating direct comparisons of immune history and clinical outcomes among cases with each lineage.

Among BA.4/BA.5 cases, 15.8% had not received any COVID-19 vaccine doses, while 2.5%, 23.6%, 48.5%, and 9.6% had received 1, 2, 3, and ≥4 doses, respectively, before their diagnosis (Table 2). Among BA.2 cases, 16.1%, 2.4%, 25.0%, 50.0%, and 6.6% had received 0, 1, 2, 3, and ≥4 doses, respectively. In logistic regression analyses adjusting for all measured covariates among cases, including calendar time (measured as the week or weekend of diagnosis), adjusted odds of having received 3 and ≥4 COVID-19 vaccine doses were 1.15 (95% confidence interval: 1.09–1.21) and 1.38 (1.27–1.49) fold higher among BA.4/BA.5 cases than BA.2 cases. As compared to 5.3% of BA.4/BA.5 cases, 3.1% of BA.2 cases had documentation of a prior SARS-CoV-2 infection. Adjusted odds of documented prior infection were 1.55 (1.43–1.69) fold higher among BA.4/BA.5 cases than among contemporaneous BA.2 cases. Accounting for potential "hybrid" immunity resulting from both prior infection and vaccination, adjusted odds of both prior documented infection and prior receipt of 1, 2, 3, and ≥4 vaccine doses were 1.70 (1.47–1.96), 1.61 (1.45–1.80), 1.78 (1.60–1.98), and 2.14 (1.89–2.42) fold higher among BA.4/BA.5 cases as compared to BA.2 cases (Table S4). Taken together, these findings suggested BA.4/BA.5 infections occurred among individuals with greater degrees of immune protection against SARS-CoV-2 than time-matched BA.2 infections.

Following a positive outpatient test, crude 30-day incidence of ED presentation, any inpatient admission, and inpatient admission associated with acute respiratory infection (ARI) diagnoses was 24.2, 3.3, and 1.3 per 1000 cases with BA.4/BA.5 infection, respectively (Fig. 2; Table 3). Similarly, for cases with BA.2 infection, crude 30-day incidence of the same outcomes was 26.4, 3.4, and 1.4 per 1000 cases. Higher-acuity endpoints of intensive care unit (ICU) admission, mechanical ventilation, and death were rare within the sample. Crude incidence of ICU admission, mechanical ventilation, and mortality per 10,000 cases over the first 30 days after diagnosis was 3.7, 1.0, and 0.8 among BA.4/BA.5 cases and 3.4, 0.9, and 1.3 among BA.2 cases, respectively. In analyses restricted to cases eligible for follow-up of ≥60 days, crude incidence of ICU admission, mechanical ventilation, and death was 5.3, 1.1, and 1.3 per 10,000 BA.4/BA.5 cases, and 6.8, 1.3, and 3.9 per 10,000 BA.2 cases, respectively.

After adjustment for calendar time as well as clinical and epidemiologic factors listed in Tables 1 and 2, we did not identify statistically-significant, independent associations of BA.4/BA.5 lineage infection with risk of any of the studied clinical outcomes (Table 3). Compared to observations among BA.2 cases, adjusted hazards of ED presentation and hospital admission among BA.4/BA.5 cases were 0.96 (0.87–1.06) and 0.96 (0.73–1.26) fold as high over 30 days following diagnosis. Likewise, adjusted hazards of ED presentation and hospital admission were 0.95 (0.84–1.07) and 0.96 (0.73–1.27) fold as high among BA.4/BA.5 cases as compared to BA.2 cases over the first 15 days after diagnosis (Table S5), a period during which such outcomes have

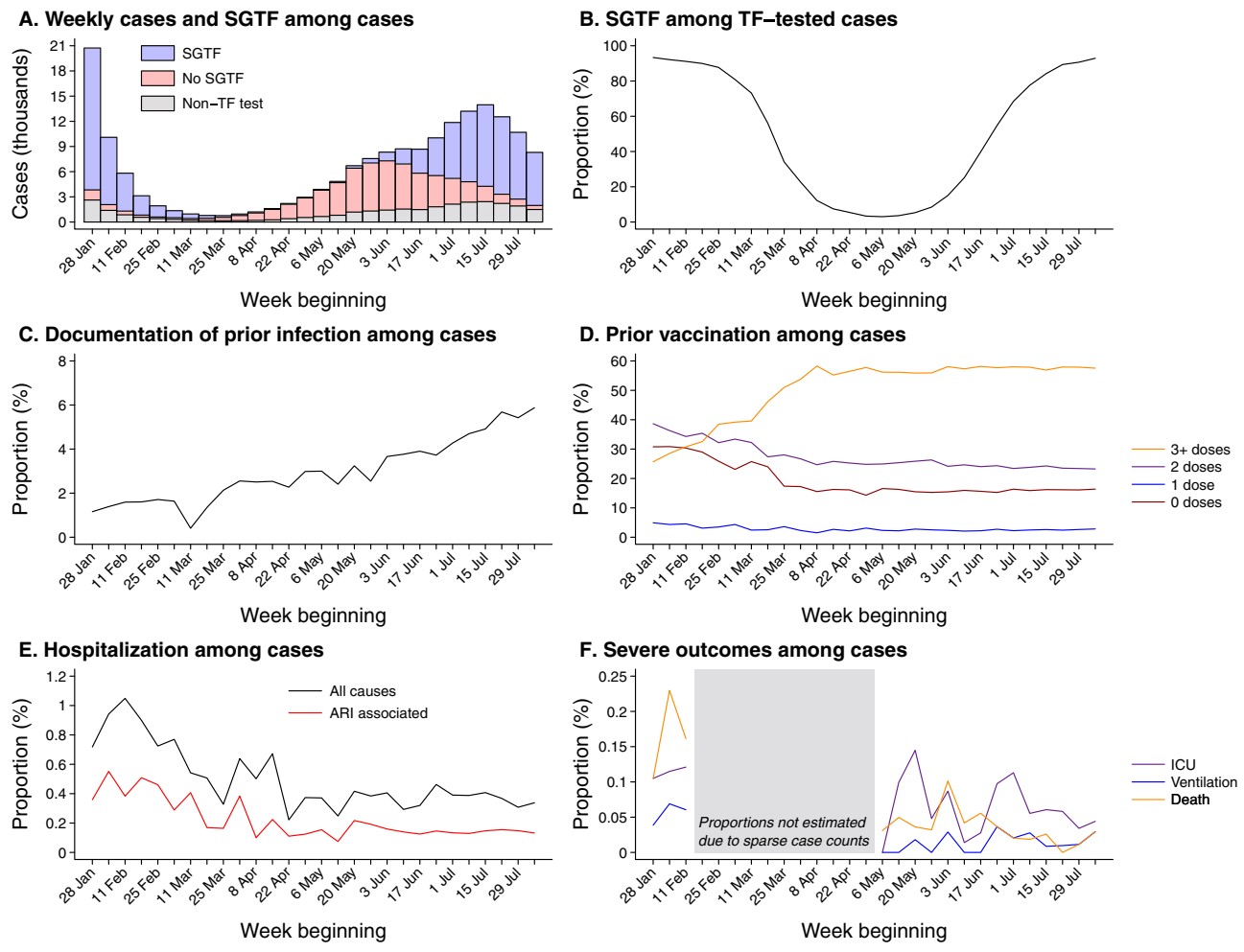

**Fig. 1 | Attributes and clinical outcomes among cases diagnosed in outpatient settings.** We first illustrate total outpatient-diagnosed cases, distinguishing those not tested using the TaqPath ThermoFisher COVID-19 Combo Kit (TF) assay and those determined to exhibit or not exhibit S gene target failure (SGTF; **A**). All subsequent plots are restricted to the eligible sample of outpatient cases tested using the TF assay, including the proportion of cases exhibiting SGTF (**B**); the proportion of cases with a history of prior documented infection (**C**); the proportion of cases who previously received 0, 1, 2, or ≥3 COVID-19 vaccine doses (**D**); the proportion of cases hospitalized within 30 days following their positive test (**E**), and the proportion of cases experiencing severe outcomes of intensive care unit (ICU) admission, mechanical ventilation, or death within 60 days after their positive test (**F**). The gray shaded area in **F** delineates weeks with <3000 cases, precluding reliable estimation of rare endpoints. Data encompass 148,105 cases tested during the study period, among whom 106,532 were tested using the TF assay and included in primary analyses.

greater specificity as markers of COVID-19 progression[16,17]. Consistent with this observation, adjusted hazards ratios (aHRs) comparing BA.4/BA.5 cases to BA.2 cases were 1.09 (0.70–1.69) for ARI-associated hospital admission over 30 days after diagnosis, and 0.68 (0.36–1.27) for ICU admission over 60 days after diagnosis (Table 3). Instances of mechanical ventilation and death were too infrequent within the sample to support multivariate regression analyses adjusting for potential confounding factors. Collectively, these findings suggest that infection with the BA.4/BA.5 or BA.2 lineages was not independently associated with clinically-meaningful differences in risk of severe outcomes among cases within the study cohort.

Within these analyses, prior COVID-19 vaccination remained independently associated with protection against progression to ED presentation, hospital admission, ARI-associated hospital admission, and ICU admission for both BA.4/BA.5 and BA.2 cases (Table 4 and Supplementary Tables S6, S7). Effect size estimates for associations of prior COVID-19 vaccination and documented prior infection with each clinical outcome did not differ appreciably for BA.4/BA.5 cases and BA.2 cases. While our case-only analysis approach did not provide a framework for estimating protection against acquisition of BA.4/BA.5 or BA.2 infections, the lower prevalence of prior documented infection

among cases as compared to the general population of KPSC members is consistent with protective effects of infection-derived immunity (Table S8).

Because underdiagnosis of mild or asymptomatic infections could hinder our ability to control for individuals' history of SARS-CoV-2 infection, we repeated these analyses within the subgroup of 4597 cases (3139 and 1458 with BA.4/BA.5 and BA.2 infections, respectively) with documented history of SARS-CoV-2 infection before their index test during the study period. Within this analysis, adjusted hazards of emergency department presentation over 15 and 30 days following diagnosis were 0.81 (0.43–1.52) and 0.87 (0.54–1.41) fold as high among BA.4/BA.5 cases as among BA.2 cases, while adjusted hazards of hospital admission over 30 days were 1.45 (0.29–7.14) fold as high (Table S9), confirming our earlier findings that infecting variant did not independently predict differential risk of severe clinical outcomes. However, event counts were prohibitively low for adjusted analyses of ARI-associated hospital admission and higher-acuity outcomes such as ICU admission within this subgroup.

To overcome these limitations to statistical power, we further undertook risk-of-bias analyses allowing for differential degrees of under-detection of prior SARS-CoV-2 infection among individuals who

## Table 1 | Characteristics of cases with BA.2 and BA.4/BA.5 lineage SARS-CoV-2 infection

| Characteristic | n (%) BA.2 (No SGTF) | BA.4/ BA.5 (SGTF) |
|---|---|---|
| | N = 46,976 | N = 59,556 |
| Age[a] | | |
| 0–9 years | 3117 (6.6) | 3883 (6.5) |
| 10–19 years | 4312 (9.2) | 4755 (8.0) |
| 20–29 years | 5146 (11.0) | 6968 (11.7) |
| 30–39 years | 8495 (18.1) | 11324 (19.0) |
| 40–49 years | 8857 (18.9) | 11388 (19.1) |
| 50–59 years | 7952 (16.9) | 10016 (16.8) |
| 60–69 years | 5542 (11.8) | 6864 (11.5) |
| 70–79 years | 2763 (5.9) | 3371 (5.7) |
| ≥80 years | 792 (1.7) | 987 (1.7) |
| Sex | | |
| Female | 25,905 (55.1) | 32,302 (54.2) |
| Male | 21,071 (44.9) | 27,254 (45.8) |
| Race/ethnicity | | |
| White, non-Hispanic | 11,381 (24.2) | 12,718 (21.4) |
| Black, non-Hispanic | 3276 (7.0) | 4490 (7.5) |
| Hispanic | 21,145 (45.0) | 29,489 (49.5) |
| Asian | 7503 (16.0) | 88381 (14.1) |
| Pacific Islander | 481 (1.0) | 515 (0.9) |
| Other, mixed race, or unknown race | 3190 (6.8) | 3963 (6.7) |
| Neighborhood deprivation index[a] | | |
| Quintile 1 (least deprived) | 7255 (15.4) | 7748 (13.0) |
| Quintile 2 | 10,836 (23.1) | 12,645 (21.2) |
| Quintile 3 | 12,079 (25.7) | 15,326 (25.7) |
| Quintile 4 | 10,513 (22.4) | 14,397 (24.2) |
| Quintile 5 (most deprived) | 6293 (13.4) | 9440 (15.9) |
| Cigarette smoking[a] | | |
| Never smoker | 36,935 (78.6) | 46,195 (77.6) |
| Current smoker | 2217 (4.7) | 3079 (5.2) |
| Former smoker | 7824 (16.7) | 10,282 (17.3) |
| Body mass index[a] | | |
| Underweight (<18.5) | 3992 (8.5) | 4866 (8.2) |
| Normal weight (18.5–24.9) | 11,983 (25.5) | 14,383 (24.2) |
| Overweight (25.0–29.9) | 14,203 (30.2) | 18,108 (30.4) |
| Obese (30–39.9) | 13,072 (27.8) | 17,257 (29.0) |
| Morbidly obese (≥40) | 3726 (7.9) | 4942 (8.3) |
| Charlson comorbidity index | | |
| 0 | 34,661 (73.8) | 43,825 (73.6) |
| 1–2 | 9791 (20.8) | 12,460 (20.9) |
| 3–5 | 1937 (4.1) | 2510 (4.2) |
| ≥6 | 597 (1.3) | 761 (1.3) |
| Prior year outpatient visits | | |
| 0–4 | 13,281 (28.3) | 18,689 (31.4) |
| 5–9 | 12,924 (27.5) | 16,595 (27.9) |
| 10–14 | 7546 (16.1) | 894 (15.1) |
| 15–19 | 4375 (9.3) | 5262 (8.8) |
| 20–29 | 4582 (9.8) | 5287 (8.9) |
| ≥30 | 4268 (9.1) | 4729 (7.9) |
| Prior year ED visits | | |
| 0 | 39,281 (83.6) | 49,909 (83.8) |
| 1 | 5616 (12.0) | 7193 (12.1) |
| 2 | 1315 (2.8) | 1580 (2.7) |
| ≥3 | 764 (1.6) | 874 (1.5) |
| Prior year inpatient admissions | | |
| 0 | 45,012 (95.8) | 57,288 (96.2) |
| 1 | 1715 (3.7) | 1946 (3.3) |

## Table 1 (continued) | Characteristics of cases with BA.2 and BA.4/BA.5 lineage SARS-CoV-2 infection

| Characteristic | n (%) BA.2 (No SGTF) | BA.4/ BA.5 (SGTF) |
|---|---|---|
| 2 | 176 (0.4) | 231 (0.4) |
| ≥3 | 73 (0.2) | 91 (0.2) |
| Receipt of other vaccinations | | |
| 2021–22 season influenza vaccine | 26,704 (56.8) | 33,082 (55.5) |
| 13-valent pneumococcal conjugate vaccine | 10,677 (22.7) | 13,058 (21.9) |
| 23-valent pneumococcal polysaccharide vaccine | 12,477 (26.6) | 15,990 (26.8) |
| Receipt of nirmatrelvir-ritonavir[b] | | |
| Not received within 14 days of diagnosis | 44,550 (94.8) | 56,149 (94.3) |
| Received within 14 days of diagnosis | 2426 (5.2) | 3407 (5.7) |

SGTF: S gene target failure, here interpreted as a proxy for SARS-CoV-2 variant; CI Confidence interval.
[a]Multiple imputation was used to address missing data; numbers may not add to column totals where missing values occur. The number of missing observations is as follows for the indicated covariates: age: 1 (<0.1%) BA.4/BA.5 cases, 0 (0.0%) BA.2 cases; neighborhood deprivation index: 43 (0.1%) BA.4/BA.5 cases, 38 (0.1%) BA.2 cases; cigarette smoking: 9600 (16.1%) BA.4/BA.5 cases, 7174 (15.3%) BA.2 cases; BMI: 10,917 (18.3%) BA.4/BA.5 cases, 7616 (16.2%) BA.2 cases.
[b]Logistic regression analyses reported in Table 2 did not include receipt of nirmatrelvir-ritonavir as a covariate predicting infecting lineage. Cox proportional hazards models reported in Tables 3, 4, Supplementary Tables S5–S7 censored at timing of first nirmatrelvir-ritonavir dispense.

ultimately experienced, or did not experience, each clinical outcome, consistent with the framework of prior analyses[4]. Across the range of parameters considered, we did not identify scenarios where 95% confidence intervals would rule out the null hypothesis of equivalent risk of hospital admission for any cause, ARI-associated hospital admission, or ICU admission among BA.4/BA.5 and BA.2 cases (Fig. S1). Differences in risk of ED presentation over 15 days and 30 days were expected to meet this threshold of statistical significance only when true prevalence of prior infection was modeled as ≥3-fold higher-than-observed among cases who ultimately presented to the ED, and ≥9-fold higher-than-observed among cases who did not. However, even under this scenario, bias-corrected effect sizes were expected to convey only modest differences in risk among BA.4/BA.5 and BA.2 cases (aHRs equal to 1.21 [1.07–1.37] and 1.22 [1.10–1.35], respectively, for ED presentation over 15 and 30 days among BA.4/BA.5 cases as compared to BA.2 cases). Thus, these analyses supported our primary findings that infecting lineage was unlikely to be an independent predictor of clinically-meaningful differences in risk for severe outcomes.

## Discussion

Our analysis has provided insight into several characteristics of SARS-CoV-2 BA.4/BA.5 Omicron lineage infections. First, outpatient-diagnosed BA.4/BA.5 cases had at least 55% higher adjusted odds of a prior documented infection (primarily with pre-Omicron variants) than time-matched BA.2 cases, as well as modestly higher adjusted odds of having received ≥3 COVID-19 vaccine doses. These findings corroborate earlier suggestions of immune escape in BA.4/BA.5 infections, which to date have been based largely on data from genomic[10] and neutralization[11,12,18] analyses rather than direct clinical evidence. Reassuringly, however, our findings are consistent with previous evidence that vaccination remains protective against severe disease associated with the BA.4/BA.5 lineages, at levels comparable to those reported for the BA.2 lineage[5,19–21]. Within our large sample of 49,976 BA.2 cases and 59,556 BA.4/BA.5 cases, vaccination was not associated with statistically meaningful differences in estimates of protection against progression from an initial outpatient diagnosis to subsequent illness requiring ED presentation or either hospital or ICU admission. As our study is limited to infected cases who received clinical molecular testing, it is important to note our findings do not

**Table 2 | Prior vaccination and documented SARS-CoV-2 infection among cases with BA.2 and BA.4/BA.5 lineage SARS-CoV-2 infection**

| Characteristic | n (%) | | OR (95% CI) | |
|---|---|---|---|---|
| | BA.2 (No SGTF) | BA.4/BA.5 (SGTF) | Unadjusted[c] | Adjusted[c] |
| | N = 46,976 | N = 59,556 | | |
| Vaccination—doses received[a] | | | | |
| 0 doses | 7543 (16.1) | 9422 (15.8) | ref. | ref. |
| 1 dose | 1121 (2.4) | 1503 (2.5) | 1.08 (0.97, 1.21) | 1.09 (0.98, 1.23) |
| 2 doses | 11,743 (25.0) | 14,038 (23.6) | 1.02 (0.97, 1.07) | 1.04 (0.98, 1.10) |
| 3 doses | 23,477 (50.0) | 28,902 (48.5) | 1.08 (1.03, 1.13) | 1.15 (1.09, 1.21) |
| ≥4 doses | 3092 (6.6) | 5691 (9.6) | 1.14 (1.06, 1.22) | 1.38 (1.27, 1.49) |
| Prior infection (≥90 days before positive test)[b] | | | | |
| None documented | 45,518 (96.9) | 56,417 (94.7) | ref. | ref. |
| Any documented | 1458 (3.1) | 3139 (5.3) | 1.52 (1.40, 1.65) | 1.55 (1.43, 1.69) |

*SGTF* S gene target failure, here interpreted as a proxy for SARS-CoV-2 lineage, *CI* Confidence interval, *OR* Odds ratio.
[a]Vaccine doses received are summed across all products.
[b]We present estimates stratified by both vaccination and prior documented infection (hybrid immunity) in Table S4. We present analyses subset to cases with documented prior infection, and exploring bias resulting from potential misclassification of prior infection status, in Table S8 and Fig. S1.
[c]Odds ratios and adjusted odds ratios are estimated using logistic regression models defining cases' calendar week (or weekend) of diagnosis as strata. Adjusted estimates control for all variables listed in Table 1 as covariates with the exception of nirmatrelvir-ritonavir receipt.

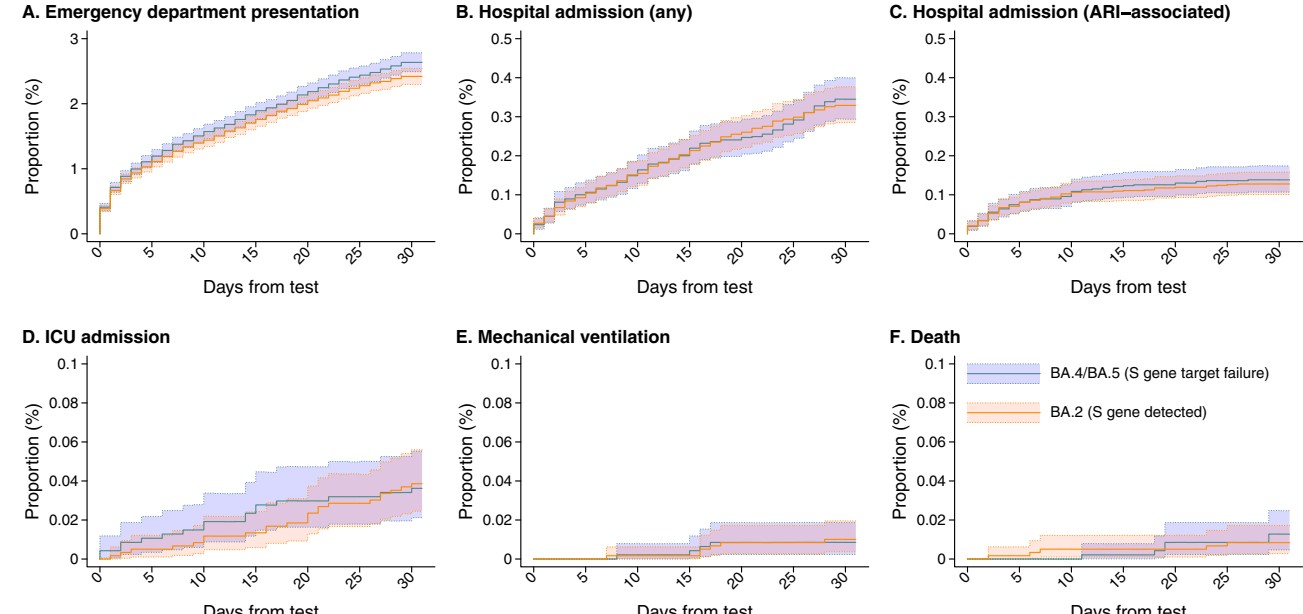

**Fig. 2 | Clinical outcomes among cases with BA.2 and BA.4/BA.5 lineage SARS-CoV-2 infection, tested 29 April, 2022 to 29 July, 2022.** Plots illustrate cumulative 30-day risk of severe clinical outcomes among cases first ascertained in outpatient settings, stratified by SGTF status for infecting subvariant (BA.4/BA.5 [SGTF]: orange; BA.2 [No SGTF]: blue), for endpoints of any emergency department (ED) presentation (**A**); any inpatient admission (**B**); inpatient admission associated with an acute respiratory infection (ARI) diagnosis (**C**); intensive care unit (ICU) admission (**D**); mechanical ventilation (**E**), and death (**F**). Shaded areas denote 95% confidence intervals around median estimates (center lines). Plotted estimates indicate absolute risk of each outcome and do not include adjustment for confounding differences between cases with BA.2 and BA.4/BA.5 infection. Adjusted hazards ratios presented in Table 3 should thus be interpreted as measures of the independent association of infecting lineage with risk of each outcome. Data encompass outcomes among 106,532 SARS-CoV-2 cases (49,976 with BA.2 infections and 59,556 with BA.4/BA.5 infections).

measure the effectiveness of prior infection or vaccination against infection with either lineage BA.4/BA.5 or BA.2 lineages. However, at least one previous study has further demonstrated that prior infection, especially with BA.1 or BA.2 Omicron lineages, remains modestly protective against BA.4/BA.5 infection, although at lower levels than those seen for earlier Omicron lineages and pre-Omicron variants[22].

We also identify that BA.4/BA.5 infections were not associated with enhanced risk of subsequent healthcare utilization indicative of disease progression, including ED presentation, hospital admission, or other severe endpoints, relative to BA.2 infections. As we have established in prior work that BA.2 and BA.1 lineage infections likewise do not differ in clinical severity within the KSPC population[4], our findings suggest that reductions in the severity of disease caused by the BA.1 Omicron lineage, relative to the Delta variant, have persisted with BA.4/BA.5. While estimates of the severity of illness associated with BA.4/BA.5 lineage infections remain limited, our findings are consistent with those of several other studies. During the first weeks following BA.4/BA.5 emergence in South Africa, BA.4/BA.5 infections did

**Table 3 | Association of infecting lineage with risk of severe clinical outcomes among cases tested 29 April, 2022 to 29 July, 2022**

| Clinical endpoint | Infecting lineage | n (%)ᵃ | Events | Hazard ratio (95% CI) | |
|---|---|---|---|---|---|
| | | | Rate per 100,000 person-days | Unadjusted | Adjusted |
| Emergency department presentation—30 days | | | | | |
| | BA.2 (*S gene detected*) | 1238 (2.6) | 93.2 | ref. | ref. |
| | BA.4/BA.5 (*S gene target failure*) | 1441 (2.4) | 86.7 | 0.91 (0.82, 1.01) | 0.96 (0.87, 1.06) |
| Hospital admission—30 days | | | | | |
| | BA.2 (*S gene detected*) | 162 (0.34) | 11.8 | ref. | ref. |
| | BA.4/BA.5 (*S gene target failure*) | 196 (0.33) | 11.6 | 0.92 (0.70, 1.21) | 0.96 (0.73, 1.26) |
| ARI-associated hospital admission—30 days | | | | | |
| | BA.2 (*S gene detected*) | 65 (0.13) | 4.7 | ref. | ref. |
| | BA.4/BA.5 (*S gene target failure*) | 76 (0.13) | 4.5 | 1.02 (0.65, 1.60) | 1.09 (0.70, 1.69) |
| Intensive care unit admission—60 days | | | | | |
| | BA.2 (*S gene detected*) | 29 (0.068) | 1.2 | ref. | ref. |
| | BA.4/BA.5 (*S gene target failure*) | 13 (0.049) | 1.1 | 0.62 (0.33, 1.17) | 0.68 (0.36, 1.27) |
| Mechanical ventilation—60 days | | | | | |
| | BA.2 (*S gene detected*) | 5 (0.012) | 0.2 | – | – |
| | BA.4/BA.5 (*S gene target failure*) | 4 (0.015) | 0.3 | – | – |
| All-cause mortality—60 days | | | | | |
| | BA.2 (*S gene detected*) | 18 (0.068) | 0.7 | – | – |
| | BA.4/BA.5 (*S gene target failure*) | 4 (0.0094) | 0.3 | – | – |

*CI* Confidence interval. Estimates indicate the adjusted hazard ratios (aHR) of each outcome, comparing cases with BA.4/BA.5 infection to those with BA.2 infection, estimated via Cox proportional hazards models including strata for cases' week of diagnosis and all covariates listed in Tables 1 and 2.
ᵃProportions calculated among 46,976 BA.2 cases and 59,556 BA.4/BA.5 cases followed ≥30 days (for endpoints of emergency department presentation and hospital admission), and among 42,746 BA.2 cases and 26,339 BA.4/BA.5 cases followed ≥60 days (for endpoints of intensive care unit admission, mechanical ventilation, and all-cause mortality).

not differ in severity from BA.1 infections, although statistical power in these analyses was constrained (*n* = 1806 BA.4/BA.5 cases analyzed) and data on cases' clinical comorbidities and healthcare-seeking behavior were not available to fully support causal inference addressing the role of infecting variant[9]. Consistent with this finding, risk of hospital admission during the BA.4/BA.5 and BA.1 waves in South Africa did not differ within analyses of all diagnosed cases[23]. Whereas a population-based study in Denmark suggested moderately increased risk of hospital admission among BA.5 cases as compared to BA.2 cases[21], this analysis did not include adjustment for potentially relevant confounders including individuals' healthcare-seeking behavior and calendar time. Our analyses sought to adjust for these variables based on the observed association of prior vaccination or infection with heightened risk of BA.4/BA.5 breakthrough infection, and because patient and provider demand for clinical SARS-CoV-2 testing changed markedly over the course of the BA.2 and BA.4/BA.5 waves, as public health mitigation measures were relaxed and access to home antigen testing expanded. Emergence of BA.5 was not associated with increased burden in hospital settings within Denmark, consistent with our findings and contrary to associations reported at the level of individual cases, where differing sources of confounding may apply[24].

Our analysis has several limitations. As our sample is restricted to individuals receiving outpatient molecular testing, our findings do not convey the comprehensive burden of the BA.4/BA.5 and BA.2 lineages in the KPSC healthcare system, including cases who were admitted upon their initial testing presentation. Rather, this analytic framework enabled us to maximize internal validity for our primary study questions comparing BA.4/BA.5 and BA.2 cases with similar healthcare-

seeking behavior, and from a similar point in their disease progression. Prior infections are likely undercounted among both BA.4/BA.5 and BA.2 cases. Because this misclassification may obscure the true magnitude of differences in prevalence of prior infection among cases acquiring each lineage, the increase in odds of prior infection among BA.4/BA.5 cases as compared to BA.2 cases likely exceeds our estimate of 55%. However, our findings of equivalent risk of severe clinical outcomes with each lineage are unlikely to be driven by this factor alone. Sensitivity analyses identified BA.4/BA.5 lineage infections would be associated with higher risk of ED presentation only under extreme scenarios where ≤1 in 9 prior infections were recorded among cases who did not require ED care (representing 97.5% of cases analyzed). Even under such a scenario, bias-corrected adjusted hazards ratio estimates were expected to identify only ~20% higher risk of ED presentation among BA.4/BA.5 cases as compared to BA.2 cases; bias-corrected differences in risk of other endpoints did not reach statistical significance even within our large sample of 106,532 cases. While the true prevalence of unascertained prior infection in this population is not precisely known, this scenario represents a considerable departure from prior estimates of the reporting fraction in California[25]. Clinically-meaningful, independent associations of the BA.2 and BA.4/BA.5 lineages with risk of severe outcomes are thus unlikely. Studies employing prospective serological sampling or recruiting frequently-tested populations will be of importance for characterizing how infection-derived immunity, including from Omicron variant infections, and "hybrid" immunity from prior vaccination and infection[26], influence susceptibility to infection and disease as SARS-CoV-2 lineages continue to evolve[27]. It is important to note that our case-only

**Table 4 | Association of prior vaccination or infection with risk of severe clinical outcomes among cases tested 29 April, 2022 to 29 July, 2022**

| Population | Characteristic | Adjusted hazard ratio (95% CI), by clinical endpoint[a] | | | |
|---|---|---|---|---|---|
| | | All-cause ED presentation (30 days) | All-cause hospital admission (30 days) | ARI-associated hospital admission (30 days) | ICU admission (60 days) |
| All cases | | | | | |
| | 0 vaccine doses | ref. | ref. | ref. | ref. |
| | 2 vaccine doses | 0.85 (0.76, 0.90) | 0.86 (0.71, 1.04) | 0.62 (0.50, 1.01) | 0.77 (0.44, 1.06) |
| | 3 vaccine doses | 0.70 (0.66, 0.79) | 0.57 (0.45, 0.70) | 0.44 (0.26, 0.57) | 0.50 (0.32, 0.81) |
| | ≥4 vaccine doses | 0.67 (0.62, 0.75) | 0.55 (0.43, 0.66) | 0.35 (0.21, 0.47) | 0.28 (0.22, 0.48) |
| | No documented prior infection | ref. | ref. | ref. | ref. |
| | Any documented prior infection | 0.87 (0.77, 1.02) | 0.93 (0.69, 1.19) | 0.71 (0.45, 1.78) | 0.18 (0.09, 1.14) |
| BA.4/BA.5 cases | | | | | |
| | 0 vaccine doses | ref. | ref. | ref. | ref. |
| | 2 vaccine doses | 0.85 (0.73, 0.92) | 0.92 (0.70, 1.21) | 0.62 (0.46, 1.28) | 0.42 (0.16, 0.71) |
| | 3 vaccine doses | 0.66 (0.61, 0.77) | 0.62 (0.44, 0.82) | 0.57 (0.27, 0.81) | 0.35 (0.17, 0.76) |
| | ≥4 vaccine doses | 0.63 (0.56, 0.73) | 0.59 (0.42, 0.77) | 0.29 (0.15, 0.44) | 0.27 (0.21, 0.52) |
| | No documented prior infection | ref. | ref. | ref. | – |
| | Any documented prior infection | 0.88 (0.75, 1.07) | 0.98 (0.66, 1.35) | 0.44 (0.22, 1.77) | – |
| BA.2 cases | | | | | |
| | 0 vaccine doses | ref. | ref. | ref. | ref. |
| | 2 vaccine doses | 0.86 (0.73, 0.94) | 0.80 (0.61, 1.05) | 0.58 (0.44, 1.13) | 0.80 (0.39, 1.20) |
| | 3 vaccine doses | 0.75 (0.69, 0.89) | 0.53 (0.38, 0.70) | 0.35 (0.16, 0.50) | 0.56 (0.31, 1.05) |
| | ≥4 vaccine doses | 0.69 (0.61, 0.83) | 0.53 (0.36, 0.71) | 0.42 (0.20, 0.65) | 0.08 (0.05, 0.35) |
| | No documented prior infection | ref. | ref. | ref. | ref. |
| | Any documented prior infection | 0.87 (0.71, 1.13) | 0.93 (0.57, 1.39) | 0.85 (0.46, 2.98) | 0.31 (0.15, 1.94) |

*CI* Confidence interval. Estimates indicate the adjusted hazard ratios (aHR) of each outcome, comparing cases with BA.4/BA.5 infection to those with BA.2 infection, estimated via Cox proportional hazards models including strata for cases' week of diagnosis and all covariates listed in Table 1.
[a]Previous infection defined by any positive test result or diagnosis ≥90 days prior to the date of the current test. We omit estimates among recipients of single vaccine doses due to sparse sample sizes (*N* = 1121 BA.4/BA.5 cases and 1503 BA.2 cases). Sample sizes for all exposure groups are presented in Table 2. Corresponding unadjusted hazard ratio estimates are presented in Table S6; estimates for 15-day monitoring for ED presentation and hospital admission are presented in Table S7.

approach compares characteristics (including prevalence of prior vaccination or infection) among cases with BA.4/BA.5 and BA.2 infections, but does not directly measure the effectiveness of either of these factors in preventing acquisition of infection. Prior studies, including those using serological data or recurrent testing to assign prior infections with greater accuracy, have provided important insight into protection against infection, including protection associated with prior exposure to BA.1 or pre-Omicron variants[22,28,29]. Last, our analyses do not distinguish cases infected with BA.4 and BA.5, or cases with BA.2.12.1 versus other BA.2 sublineages, which may be associated with distinct epidemiologic and clinical characteristics.

While our analyses do not distinguish specific causes of ED presentations and hospital admission, our findings hold in analyses restricted to ED presentations and hospital admissions occurring within 15 days of cases' first positive test, and hospital admissions for which ARI-related diagnosis codes were assigned, both of which are likely to have greater specificity for indicating healthcare interactions precipitated by COVID-19 illness[16,17]. As SARS-CoV-2 infection substantially increases individuals' likelihood of requiring care in higher-acuity settings, a majority of presentations and admissions occurring in the aftermath of a positive SARS-CoV-2 outpatient test were expected to be attributable to COVID-19 (Table S10)[4,30]. In contrast, incidental admissions (estimated to account for 20–23% of hospitalizations among patients infected with the Omicron variant[31–33]) could

be over-represented in analyses which included results of all SARS-CoV-2 screening tests administered to admitted patients.

Reasons for the greater degree of escape of BA.4/BA.5 from naturally-acquired immunity, as compared to vaccine-derived immunity, merit consideration to inform future updates to the design of COVID-19 vaccines. Whereas mutations in the S protein encoded by BA.4/BA.5[10] may explain the reduced effectiveness of the BNT162b2, mRNA-1273, and Ad.26.COV2.S vaccines in preventing infection with these lineages, responses to other SARS-CoV-2 surface antigens likely play a role in protection associated with prior natural infection[34,35]. Thus, mutations affecting non-S antigens of SARS-CoV-2 may account for the over-representation of BA.4/BA.5 cases with documented prior infection or hybrid immunity from infection and vaccination, as compared to findings with respect to vaccination alone.

While it is encouraging that we find BA.4/BA.5 lineage infections are not associated with differential severity in comparison to other Omicron lineages, it is important to note that disease burden is influenced by additional variant-specific properties including the intrinsic capacity to transmit and to infect individuals with immunity from prior vaccination or infection[36]. These fitness advantages are relevant to consider with BA.4/BA.5, which outcompeted BA.2 in the context of substantial population immunity[10], and with subsequent lineages replacing BA.4/BA.5 including XBB/XBB.1.5. As new SARS-CoV-2 variants continue to emerge, associations of novel circulating lineages

with risk of severe illness and post-vaccination breakthrough infection should inform public health responses.

## Methods

### Setting, procedures, and study population

Care delivery within KPSC has been described previously[4]. Briefly, approximately 19% of the population of southern California receives care from KPSC through employer-provided, pre-paid, or federally sponsored insurance plans. In-network care delivery data encompassing diagnoses (and accompanying clinical notes), immunizations, laboratory tests administered and test results, and prescriptions are captured in near-real time via patient EHRs, while out-of-network care is captured through insurance claim reimbursements. Delivery of COVID-19 vaccine doses by other providers was identified via linkage to California Immunization Registry data and other health systems using the Epic EHR system. Online portals provided an automated platform for individuals to upload or notify providers of positive at-home test results or test results received from other providers. The study protocol was approved by the KPSC Institutional Review Board.

Molecular diagnostic testing for COVID-19 was made available to all individuals receiving outpatient care from KSPC for any indication during the study period. Consistent with prior analyses[4], we restricted our analytic sample to cases with a positive SARS-CoV-2 test result from testing undertaken in outpatient settings using the ThermoFisher TaqPath COVID-19 Combo Kit (the most commonly used assay for outpatient testing at KPSC during the study period), with ≥1 year of continuous enrollment in KPSC health plans prior to their test date. We restricted analyses to cases diagnosed between 29 April to 29 July, 2022, without a prior positive test result in the preceding 90 days or at any time in 2022, to avoid dual counting and to distinguish new-onset infections from lingering positive detections of infections with earlier Omicron (e.g., BA.1) or non-Omicron (e.g., Delta) lineages, for which SGTF may not align with BA.4/BA.5 and BA.2 lineage determination. Laboratory data included qualitative (presence/absence) detection of RNA for probes targeting the SARS-CoV-2 S, nucleocapsid (N), and Orf1a/b genes. Cases with cycle threshold values below 37 for ≥2 probes were considered positive for SARS-CoV-2. As BA.4/BA.5 lineages harbor the Δ69-70 amino acid deletion in the S protein, SGTF has been proposed elsewhere as a proxy for distinguishing BA.4/BA.5 from BA.2 lineages[9,10], as supported by validation data within our study population (Tables S2 and S3). We considered cases to have SGTF if cycle threshold values were below 37 for both N and Orf1a/b, and ≥37 for S.

In addition to enabling longitudinal follow-up for severe endpoint ascertainment, restricting analyses to cases tested as outpatients was expected to provide at least three design advantages helping to mitigate bias. As a strategy to ensure sufficient hospital capacity, KPSC implemented a home-based monitoring program for high-risk outpatients diagnosed with SARS-CoV-2 infection, who were provided with medical-grade pulse oxygen monitors and called daily by healthcare providers for clinical assessment. As this program provided an opportunity to use standardized criteria for ED referral and inpatient admission, these endpoints were considered to provide internally-consistent measures of disease severity within our sample followed from the point of outpatient testing. Excluding individuals first ascertained in hospital settings further helped to reduce the potential for bias driven by differential healthcare-seeking behavior among cases tested as outpatients versus those who deferred testing to more severe stages of illness. This approach also enabled us to minimize the inclusion of cases hospitalized for other causes who were identified incidentally via SARS-CoV-2 infection screening at admission (estimated to represent roughly 20–23% of Omicron-associated admissions within the US[31,32] and other settings[33], as compared to ~12% of admissions prior to Omicron variant emergence[37]). Infections not associated with significant respiratory symptoms would be unlikely to precipitate outpatient testing, particularly during this period when home-based antigen testing was widely available. Automated text searches identified acute COVID-19 associated symptoms in the medical records of 88.8% of hospitalized patients in our sample[16]. Last, whereas inpatient admission is generally a rare event, its likelihood is greatly increased during SARS-CoV-2 infection[30]. From 2016–2019, mean annual rates of hospital admission were 7.3–8.3 per 100 person-years, respectively, among adult members of KPSC, comparable to the US population average of 7.6 per 100 person-years as of 2018[38]. Prior to Omicron variant emergence in the US, the proportion of SARS-CoV-2 infections requiring hospital admission has been estimated between 3.3% (in California) and 6.9%[39–41], equivalent to an annualized rate of 70.9–148.1 admissions per 100 person-years over the course of infection (Table S10). Thus, SARS-CoV-2 infection would be expected to increase individuals' risk of hospital admission by 9.6-fold to 20.0-fold, or by 2.8-fold to 5.9-fold when allowing for a 71% reduction in risk of hospital admission with the Omicron variant under real-world conditions of vaccination and prior infection within KPSC[42].

### Outcomes

Outcomes of interest to our analyses included: (1) any ED presentation; (2) any inpatient admission; (3) ARI-associated inpatient admission, at which physicians recorded ≥1 of the ARI diagnostic codes indicated in Table S1; (4) ICU admission; (5) mechanical ventilation; and (6) death. While use of ARI-associated diagnosis codes aimed to further limit inclusion of hospital admissions unrelated to COVID-19, this outcome may be prone to under-counting, as severe cardiovascular[43], cerebrovascular[44], and other complications of COVID-19 may not be captured. We limited follow-up time for ED presentation and hospital admission to 30 days following the initial positive outpatient test, and addressed ED presentations and hospital admissions occurring within 15 days after the initial positive outpatient test in sensitivity analyses. We included follow-up time through 60 days from the initial positive outpatient test for endpoints of ICU admission, mechanical ventilation, and death, owing to the longer course of disease expected to precede such outcomes. We censored observations at study end date or at disenrollment for cases who had not experienced each outcome. As cases diagnosed in outpatient settings were enrolled in a home-based monitoring program with standardized criteria for ED referral and inpatient admission[15], we expected severity of illness associated with each endpoint to be internally comparable within the study cohort. Last, to facilitate our ability to measure intrinsic associations of infecting lineage with risk of progression to severe outcomes, we censored observations at dates of nirmatrelvir-ritonavir dispense for individuals who received this treatment (5.5% of patients analyzed [n = 5833]). Real-world effectiveness of nirmatrelvir-ritonavir in preventing adverse clinical outcomes within this population has been described elsewhere[17,45].

### Case characteristics

We recorded the following characteristics for each case: age (defined in 10-year age bands), biological sex (as reported in patient medical records); race/ethnicity (white, black, Hispanic of any race, Asian, Pacific Islander, and other/mixed/unknown race); neighborhood deprivation index, measured at the Census block level; smoking status (current, former, or never smoker); body mass index (BMI; underweight, normal weight, overweight, obese, and morbidly obese); Charlson comorbidity index (0, 1–2, 3–5, and ≥6); prior-year emergency department visits and inpatient admissions (each defined as 0, 1, 2, or ≥3 events); prior-year outpatient visits (0–4, 5–9, 10–14, 15–19, 20–29, or ≥30 events); documented prior SARS-CoV-2 infection; and history of COVID-19 vaccination (receipt of 0, 1, 2, 3, or ≥4 doses, and time from receipt of each dose to each case's testing date), and receipt of nirmatrelvir-ritonavir ≤14 days after the initial outpatient diagnosis date. As individuals were only included at the point of their first

positive SARS-CoV-2 test during 2022, most prior documented SARS-CoV-2 infections were expected to involve pre-Omicron lineages.

## Multiple imputation of missing data

Variables with missing data included cases' age ($n = 1$; 0.00094% of 106,532 cases), neighborhood deprivation index ($n = 81$; 0.076% of cases), BMI ($n = 18,533$; 17.4% of observations), and cigarette smoking status ($n = 16,774$; 15.7% of observations). We populated 10 complete pseudo-datasets sampling from the distribution of missing values, according to the joint distribution of all measured variables, via multiple imputation, and repeated all statistical analyses across each pseudo-dataset. We pooled resulting estimates according to Rubin's rules[46].

## Logistic regression analysis

We compared the distributions of prior vaccination status and prior infection status among BA.4/BA.5 cases versus BA.2 cases via logistic regression. Models controlled for all variables listed above, with the exception of nirmatrelvir-ritonavir receipt (which occurred after diagnosis), to define aORs in relation to infecting lineage. Models included distinct intercepts for each calendar week to control for potential changes in testing and healthcare-seeking practices over the period of BA.4/BA.5 emergence.

## Survival analysis

We fit Cox proportional regression models including data from all outpatient-diagnosed cases, censoring at either the study end date, end of follow-up, disenrollment, or date of nirmatrelvir-ritonavir dispense. Models defined covariates for each case characteristic listed above. Models defined strata according to cases' calendar week of testing to control for potential changes in testing and healthcare-seeking practices over the period of BA.4/BA.5 emergence. We verified the proportional hazards assumption for all models by testing for non-zero slopes of the Schoenfeld residuals[47,48].

## Hypothesis testing

We used the two-sided $p < 0.05$ threshold to distinguish statistically-significant findings; we report 95% confidence intervals with all measures of association to convey magnitude and precision of estimates.

## Sensitivity analyses

Because protection from prior infection could contribute to lower risk of clinical progression among a higher proportion of cases with BA.4/BA.5 infection than BA.2 infection[24], we undertook several sensitivity analyses aiming to determine whether our results were robust to bias driven by potentially differential prevalence of unrecorded prior infections among cases infected with each lineage. First, we repeated our primary survival analyses within the subset of cases known to have experienced a prior infection, as differential prevalence of prior infection could not lead to differences in disease progression within this stratum. However, sample sizes were inadequate to allow similar analyses for all endpoints within this subset. Therefore, we also conducted risk-of-bias analyses allowing for non-differential or differential undercounting of prior infections among cases with BA.4/BA.5 and BA.2 infection, similar to prior work in the study population[4] and described in detail below.

Within each imputed pseudo-dataset, we fit logistic regression models to define cases' propensity for prior SARS-CoV-2 infection as a function of all measured characteristics (including SGTF status) as well as the observed occurrence of symptoms potentially associated with SARS-CoV-2[16], ED presentation, hospital admission, ICU admission, mechanical ventilation, and death. To account for potentially higher-than-observed prevalence of prior infection among all cases, we repeated analyses resampling individual infection histories at random under an assumption that true prevalence of prior infection was $\rho \in$ (1, 1.5, 2, 3) times higher than that observed based on fitted propensity scores. To further allow for potentially higher prevalence of prior infection among cases who were protected from experiencing clinical outcomes, we multiplied the estimated propensity of prior infection within these groups by a factor equal to $\alpha \times \rho$, for $\alpha \in$ (1, 1.5, 2, 3), thus allowing up to 9-fold higher-than-observed prevalence of prior infection among individuals who evaded each endpoint. We plot resulting estimates of the bias-corrected adjusted hazards ratios of each outcome, comparing BA.4/BA.5 cases to BA.2 cases in Fig. S1.

## Software

We conducted analyses using R (version 4.0.3; R Foundation for Statistical Computing, Vienna, Austria). We used the survival[49] package (version 3.5-3) for time-to-event analyses, and the Amelia II package[50] (version 1.81.1) for multiple imputation.

## Reporting summary

Further information on research design is available in the Nature Portfolio Reporting Summary linked to this article.

# Data availability

Individual-level data reported in this study are not publicly shared. Upon request and subject to review by the KPSC Institutional Review Board, KPSC may provide the de-identified aggregate-level data that support the findings of this study. De-identified data may be shared upon approval of an analysis proposal and a signed data access agreement. The corresponding authors (J.A.L., S.Y.T.) will respond to requests for data access within 14 days of receipt.

# Code availability

Analysis code is available from github.com/joelewnard/ba4ba5severity[51].

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

## Acknowledgements

This work was funded by the US Centers for Disease Control and Prevention (CDC; grant 75D3-121C11520 to S.Y.T.). The findings and conclusions in this report are those of the authors and do not necessarily represent the official position of the CDC. J.A.L. was supported by grant R01-AI14812701A1 from the National Institute for Allergy and Infectious Diseases (US National Institutes of Health), which had no role in design or conduct of the study, or the decision to submit for publication.

## Author contributions

J.A.L., V.H., and S.Y.T. contributed to the study concept and design. J.A.L., V.H., and S.Y.T. led acquisition and statistical analysis of data. J.A.L. and S.Y.T. led interpretation of data. J.A.L. drafted the manuscript. V.H., J.S.K., S.F.S., B.L., H.T., and S.Y.T. critically revised the manuscript for important intellectual content. S.Y.T. obtained funding and provided supervision.

## Competing interests

J.A.L. has received research grants and consulting honoraria unrelated to this study from Pfizer. S.Y.T. has received research grants unrelated to this study from Pfizer. The remaining authors disclose no competing interests.
