## [Peer Review File · Nature Communications]

Association of SARS-CoV-2 BA.4/BA.5 Omicron lineages with immune escape and clinical outcomeREVIEWER COMMENTS

Reviewer #1 (Remarks to the Author):

This is a well presented and well written manuscript describing the association of BA5 variant with immune escape and clinical outcome.

The study adds on to previous work from Denmark and UK among others, and these are appropriately cited and discussed.

The major findings are that BA5 in line with BA2 is associated with reduced severity relative to the delta variant and that BA5 was associated with increased risk of breakthrough infection but not with hospitalisation among previously vaccinated and unvaccinated individuals.

The weakest part of the analysis is when looking in to previous infection where only 5.3 % of BA5 cases and 3.1% of BA2 cases had a prior infection. This is clearly underdiagnosed as also mentioned by the authors. When restricting the analysis to this subgroup, table S4, there is no mentioning of the protection against infection -this should be shown. It should be more clearly stated the high likelihood of underestimation of previous infections. Previous infection should be divided by variant, as it is previously shown that protection due to previous omicron infection is better than for other variants.

Reviewer #2 (Remarks to the Author):

This paper uses data from a very large North American Health insurance database to compare the epidemiology of individuals infected with BA.2 and BA.4/5 in two key areas. Firstly, immune escape as inferred by infection rates in previously vaccinated or infected individuals and secondly, progression to severe disease given infection. The subject matter is of considerable public health importance. The analysis confirms that BA4/5 is associated with immune escape when compared to BA2 but severity is similar. The paper is very well written, the data set very large and the analysis approach is elegant. Methods are clearly written and the discussion is measured and explicitly acknowledges limitations and has appropriate public health messages. The risk of bias and subset analysis aiming to explicitly address bias add to the strength of the paper. I don't have any suggestions for improvement - a really lovely piece of work.

Reviewer #3 (Remarks to the Author):

Manuscript # NCOMMS-22-44771

Title Association of SARS-CoV-2 BA.4/BA.5 Omicron lineages with immune escape and clinical outcome

In this manuscript the authors employed a case-only approach in which they compared the vaccination status or previous infection among cases infected with the Omicron BA.4/5 and BA.2. The authors use the SGTF method the allow identification of the two different Omicron variants. The authors show that individuals with a previous infection had a higher risk of infection with Omicron BA.4/5 than with BA.2, whereas this increased risk is much lower after vaccination. However, BA.4/BA.5 infection was not associated with differential risk of emergency department presentation, hospital admission, or intensive care unit compared to BA.2 infections admission

The manuscript brings an important message and contains convincing data of a large study population. There are points that need some attention.

Comments and suggestions

- SGTF and the predictive value thereof is only mentioned in the methods section, although explanation of it and the predictive value of it is important to address in the introduction or results section. (see also specific comments for the methods section)
- In general the manuscript is well written. However due to the high data density, the authors

could consider adding a summarizing/conclusive sentence after several paragraphs within the Results section to help make the paper accessible for a broad audience.

- It's not clear through the manuscript how hybrid immunity is handled. Are those cases excluded for the vaccinated group? And are individuals with a previous infection not vaccinated? If not, hybrid immunity should be handled separately. It has been clearly shown that hybrid immunity is a different story compared to either vaccination or infection alone.
- The biggest effect is seen when comparing BA.4/5 and BA.2 infections in individuals with a previous infection. However the focus in the results and abstract seems to be towards the difference among individuals with 3 or 4 vaccinations, while that effect is only modest. Consider rephrasing. It seems that there is more escape from previous infection by BA.4/5 compared to BA.2 while no/little differences in escape between BA.4/5 and BA.2 after vaccination. The authors should speculate on a possible mechanism behind this difference in immunity and escape.
- The authors state that there is little to no differences between BA.4/5 and BA.2 infected individuals in severity/hospital admission. Panel A and D of Figure 2 suggest there is a significant difference. (no reference to this figure is made in the text). Also an adjusted OR of 1.09 and 0.68 suggest otherwise. It's not clear why the authors consider this as no difference. As this is one of the major messages in the paper, this needs clarification.
- The authors mention lack of insight in hospitalizations because of (due to) and with SARS-COV-2 infection. This might cause a major bias in the low level of differences observed in severity. If only minor fractions are because of COVID and the majority of infections is in patients hospitalized for other causes it is to be expected there is no difference due to different variants. What are ratios of hospitalizations due to and with COVID? (also see below in specific comments for the methods section)

Minor comments Methods:

1. Please also mention date range of the study period in the first section of the method.
2. Second Alinea, Please mention the other end-points
3. Is there data available on reason for hospital uptake/ED presentation/other end point? As outpatient positive SARS-CoV-2 test could be followed by an unrelated other event. Otherwise, if data not present please support the statement at the last 2 sentences of the second alinea with data. Including the time range following the SARS-CoV-2 infection for which the risk of events is increased in the statement.
4. Third alinea validation of SGTF. Instead of presenting the SGTF result among BA.2, BA.4 and BA.5 specimens, presenting the results as variants found within the SGTF and non-SGTF pools is more reasonable as this is the input for the model. For example BA.1 specimens may lead to an over estimation of BA.4/5 early on in the study. SO confirmation of PPV needs to be the other way around.
5. In addition the SGTF results needs to be validated by time period (if possible by week). As depending on the variant(s) circulating (non-)SGTF is a good proxy for BA.4/5 / BA.2. Did the authors use a cut-off value for Ct avlues? It is generally known that the S target is the first to become negative upon higher Ct values (lower viral load). This would lead to a false claim of SGTF, while just being a sensitivity issue for the three different targets. Therefor samples with a high Ct value need to be excluded.

Reviewer #1 (Remarks to the Author):

This is a well presented and well written manuscript describing the association of BA5 variant with immune escape and clinical outcome.

The study adds on to previous work from Denmark and UK among others, and these are appropriately cited and discussed.

The major findings are that BA5 in line with BA2 is associated with reduced severity relative to the delta variant and that BA5 was associated with increased risk of breakthrough infection but not with hospitalisation among previously vaccinated and unvaccinated individuals.

We thank the Reviewer for this thoughtful assessment of the manuscript and for the recommendations below which we believe have helped to improve the manuscript.

The weakest part of the analysis is when looking in to previous infection where only 5.3 % of BA5 cases and 3.1% of BA2 cases had a prior infection. This is clearly underdiagnosed as also mentioned by the authors. When restricting the analysis to this subgroup, table S4, there is no mentioning of the protection against infection -this should be shown. It should be more clearly stated the high likelihood of underestimation of previous infections.

We have provided further information to help clarify this point. First, it is important to note that the 5.3% and 3.1% estimates here are low not only due to under-diagnosis, as we address, but also due to selection criteria for the study: to limit confusion with regard to repeat infections versus repeated positive testing during a continuous infection, and to support interpretation of SGTF for distinguishing BA.4/BA.5 vs. BA.2, we included only the first infection detected for each individual during 2022. Thus, the sample is enriched in individuals without prior infection (including prior Omicron infections). We have thus clarified that the exposure we measure is essentially limited to documented infection with pre-Omicron variants (line 170-171 and 457-457), and have emphasized this aspect of eligibility criteria in the revised Methods section for further clarity (line 397-399).

As the primary objective of our analysis is to distinguish severity and other characteristics of BA.2 and BA.4/BA.5 infections rather than evaluate the effectiveness of vaccination or naturally-acquired immunity against acquisition of infection, our analysis compares cases with each variant. Monitoring a cohort prospectively for acquisition of infection would be required to quantify protection against infection in a precise way, and given the issues with underestimation of prior infection, such an analysis would encounter limitations in this administrative electronic health records dataset; this is thus outside the scope of our study. However, we agree with the Reviewer about the importance of emphasizing that there is, indeed, protection against infection. To show this, we have added Table S6, which compares the prevalence of prior infection within the full population of KPSC members versus among newly-diagnosed cases. We address in the text that the lower prevalence of prior infection among cases supports the protective effect of prior infection, and clarify in the Discussion that our estimates based on comparison of BA.4/BA.5 and BA.2 cases should not be taken to measure naturally-acquired immunity against acquisition of each infection (as well as the limitations of our study in addressing this specific point; lines 178-183 and 224-227).

Previous infection should be divided by variant, as it is previously shown that protection due to previous omicron infection is better than for other variants.

Several aspects of the study, in our view, limit the reliability of trying to assign individuals' prior infections to specific infecting variants. First, a majority of infections ultimately go undiagnosed, so our limited capture of all prior infections would pass forward to variant-specific infection assignments. Second, SGTF readout is available only for infections tested in outpatient settings, meaning lineages cannot be assigned for the sizeable fraction of infections diagnosed in inpatient settings. Third, SGTF readout (where available) would not be useful for distinguishing among all pre-Omicron variants. For these reasons, we do not think it is possible to make accurate assignments as to which variant individuals were exposed to previously. However, the fact that nearly all prior infection recorded in this study involves pre-Omicron variants mitigates the Reviewer's concern about differences in the nature of protection from Omicron and pre-Omicron variants. We have addressed this issue in the revised limitations paragraph and underscored the contributions of studies which have specifically compared the effects of prior infection with Omicron or pre-Omicron variants (lines 180-183 and 227-229).

Reviewer #2 (Remarks to the Author):

This paper uses data from a very large North American Health insurance database to compare the epidemiology of individuals infected with BA.2 and BA.4/5 in two key areas. Firstly, immune escape as inferred by infection rates in previously vaccinated or infected individuals and secondly, progression to severe disease given infection. The subject matter is of considerable public health importance. The analysis confirms that BA4/5 is associated with immune escape when compared to BA2 but severity is similar. The paper is very well written, the data set very large and the analysis approach is elegant. Methods are clearly written and the discussion is measured and explicitly acknowledges limitations and has appropriate public health messages. The risk of bias and subset analysis aiming to explicitly address bias add to the strength of the paper. I don't have any suggestions for improvement - a really lovely piece of work.

We thank the Reviewer for this thoughtful assessment of the manuscript and for their kind compliments.

Reviewer #3 (Remarks to the Author):

Manuscript # NCOMMS-22-44771

Title Association of SARS-CoV-2 BA.4/BA.5 Omicron lineages with immune escape and clinical outcome

In this manuscript the authors employed a case-only approach in which they compared the vaccination status or previous infection among cases infected with the Omicron BA.4/5 and BA.2. The authors use the SGTF method the allow identification of the two different Omicron variants.

The authors show that individuals with a previous infection had a higher risk of infection with Omicron BA.4/5 than with BA.2, whereas this increased risk is much lower after vaccination.

However, BA.4/BA.5 infection was not associated with differential risk of emergency department presentation, hospital admission, or intensive care unit compared to BA.2 infections.

The manuscript brings an important message and contains convincing data of a large study population. There are points that need some attention.

We thank the Reviewer for this evaluation and for the suggestions below, which have helped to improve the manuscript in this revision.

Comments and suggestions

- SGTF and the predictive value thereof is only mentioned in the methods section, although explanation of it and the predictive value of it is important to address in the introduction or results section. (see also specific comments for the methods section)

We have added these data to the first paragraph of the Results, including a specific reference to Table S2 where we have added a breakdown of predictive value by study month (per the Reviewer's final minor comment; lines 81-84).

- In general the manuscript is well written. However due to the high data density, the authors could consider adding a summarizing/conclusive sentence after several paragraphs within the Results section to help make the paper accessible for a broad audience.

We have added a concluding sentence to emphasize key points at the end of several paragraphs within the Results section (lines 83-84, 94-95, 107-109, 131-133, 148-149, and 163-165).

- It's not clear through the manuscript how hybrid immunity is handled. Are those cases excluded for the vaccinated group? And are individuals with a previous infection not vaccinated? If not, hybrid immunity should be handled separately. It has been clearly shown that hybrid immunity is a different story compared to either vaccination or infection alone.

We agree about the importance of this distinction. We have added Table S3 showing the estimated associations of SGTF with prior vaccination and documented history of infection, defining "unvaccinated individuals without documented prior infection" as the reference group to better illustrate the combined effects of vaccination and infection, and have described these Results in the main text (lines 104-107). We have clarified in the caption to Table S3 that our analysis did not identify statistically-significant interactions between prior documented infection and vaccination (i.e., the effect of prior documented infection was similar for individuals who received 0, 1, 2, 3, or ≥ 4 doses), and present measures of the interaction parameter in this caption; the same was also true for tests of interaction between prior documented infection and risk of ED presentation, hospital admission, ARI-associated hospital admission, and ICU admission (perhaps unsurprisingly, as limited power is available with low prevalence of prior documented infection). However, as our analyses could be constrained by incomplete capture of prior infections, we caution in the revised Discussion that assessments of hybrid immunity may benefit from study designs which include prospective antibody measurement or other strategies (lines 221-224).

- The biggest effect is seen when comparing BA.4/5 and BA.2 infections in individuals with a previous infection. However the focus in the results and abstract seems to be towards the difference among individuals with 3 or 4 vaccinations, while that effect is only modest. Consider rephrasing. It seems that there is more escape from previous infection by BA.4/5 compared to BA.2 while no/little differences in escape between BA.4/5 and BA.2 after vaccination. The authors should speculate on a possible mechanism behind this difference in immunity and escape.

We have rephrased the Abstract to present the difference with respect to previous infection before the difference with respect to vaccination, which we agree is smaller and should not be emphasized (lines 23-25). This is consistent with how the finding is presented in the first paragraph of the Discussion (lines 169-172). We have also added text to the Discussion explaining that the greater difference associated with prior infection, as compared to vaccination, may reflect escape involving non-Spike antibodies with BA.4/BA.5, as vaccines available to the study population confer responses only against Spike (lines 243-249) whereas these other antibodies may play an important role in protection against reinfection (and thus breakthrough after prior infection).

- The authors state that there is little to no differences between BA.4/5 and BA.2 infected individuals in severity/hospital admission. Panel A and D of Figure 2 suggest there is a significant difference. (no reference to this figure is made in the text). Also an adjusted OR of 1.09 and 0.68 suggest otherwise. It's not clear why the authors consider this as no difference. As this is one of the major messages in the paper, this needs clarification.

It is important to note when looking at Figure 2 that these panels present unadjusted risk of each outcome among BA.2 and BA.4/BA.5 cases, rather than the independent association of infecting lineage with risk of each outcome. As immune status (vaccination and prior infection) differs among cases with each lineage, and norms were changing over time with respect to clinical SARS-CoV-2 testing and treatment (including Paxlovid access/availability), we emphasize the measures of association reported in Table 3 over the unadjusted/descriptive data presented in Figure 2 as the basis for hypothesis testing. We have clarified this in the caption to Figure 2. We also thank the Reviewer for noting that Figure 2 was not called out and have added a reference to it the Results (line 113).

We have clarified that we use the threshold two-sided $p < 0.05$ threshold to distinguish findings for which there is statistically-meaningful evidence of differences in risk (lines 356-357) and have ensured that the text of the Results is consistent in distinguishing meet or do not meet the significance threshold. We do not wish to emphasize statistical hypothesis testing as the sole basis for interpretation of the data, as some outcomes are rare in our sample (e.g., the ICU endpoint for which we estimate an aHR of 0.68 [0.36-1.27] is observed among only 13 BA.4/BA.5 cases, opening the door to potential issues such as sparse data bias, as described by Greenland et al., *BMJ* 2016). However, since a clear difference cannot be made out even among >100,000 cases in the sample, any differences (if truly present) would be unlikely to have great clinical significance as well. We have revised the text of the Results and Discussion to convey this interpretation (lines 131-133, 163-165, and 217-221).

- The authors mention lack of insight in hospitalizations because of (due to) and with SARS-COV-2 infection. This might cause a major bias in the low level of differences observed in severity. If only minor fractions are because of COVID and the majority of infections is in patients hospitalized for other causes it is to be expected there is no difference due to different variants. What are ratios of hospitalizations due to and with COVID? (also see below in specific comments for the methods section)

We revised the Methods and Discussion to clarify several steps taken in our study design to ensure that our analysis focuses on hospitalizations “due to” rather than “with” COVID-19. We note that this is an issue affecting many studies with COVID-19 hospital admission endpoints in administrative data, including prior studies on this subject (e.g., Wolter et al., *Nature Communications* 2022, which used surveillance data and could not distinguish reasons for hospital admission, and Hansen et al., *Lancet Infect Dis* 2022, which relied on ICD-10 codes to define reasons for admission).

First, we have designed the analysis to deliberately exclude identified through inpatient screening by following individuals from the point of an outpatient diagnosis forward. Whereas each individual's daily likelihood of hospital admission is very low, the risk of hospital admission is substantially increased during SARS-CoV-2 infection. Thus, the daily probability of hospital admission due to SARS-CoV-2 following a positive outpatient test greatly exceeds the daily probability of admission due to other causes, meaning a vast majority of observed hospitalizations within this sample are expected to be due to COVID-19. The same could not be assumed if we had included infections identified at the point of hospital admission, since a substantial proportion of admissions could incidentally yield a positive SARS-CoV-2 result if infection is sufficiently prevalent in the community (it has estimated that ~20-23% of Omicron hospitalizations are incidental in literature we have cited in this revision; lines 233-241 and 408-430). To further substantiate this point quantitatively, we have added estimates of the incidence rate ratio of hospitalization “due to” or “with” COVID-19 in Table S8, drawing on a variety of data sources (as detailed in response to minor point #3 below).

Second, we have included a specific endpoint for hospital admissions where an acute respiratory infection-related diagnosis was recorded (as listed in Table S1), which represent an equal proportion of BA.2 and BA.4/BA.5 hospital admissions (Table 3). We have added to our description of this endpoint in the Methods section to clarify interpretation with respect to this issue (lines 432-436).

Third, we have included sensitivity analyses evaluating ED presentation and hospital admission endpoints over 15 days after diagnosis, during which a greater proportion of healthcare utilization would be expected to result from COVID-19 specifically.

Last, we have explained that KPSC enrolled high-risk outpatients in a home-based monitoring program with standardized criteria for ED referral and inpatient admission (as described in the citation to Huynh et al., Permanente J 2021, lines 408-414), which helped to ensure that unnecessary care presentations were avoided and that inpatient admission represents a valid and internally-consistent endpoint with respect to disease severity.

Minor comments Methods:

1. Please also mention date range of the study period in the first section of the method.

We have added the dates to the first section of the Methods (lines 397-398).

2. Second Alinea, Please mention the other end-points.

We have listed all endpoints in lines 432-434.

3. Is there data available on reason for hospital uptake/ED presentation/other end point? As outpatient positive SARS-CoV-2 test could be followed by an unrelated other event. Otherwise, if data not present please support the statement at the last 2 sentences of the second alinea with data. Including the time range following the SARS-CoV-2 infection for which the risk of events is increased in the statement.

There is no automated way to determine whether COVID-19 was the cause of a hospital admission or ED presentation in the EHR data collected, although we have revised the text to note that a natural language processing algorithm (previously described by Malden et al., JMIR 2022) identified descriptions of acute COVID-19 associated symptoms among 89% of hospitalized patients in this study (lines 421-422). This approach may nonetheless under-estimate the proportion of cases with medically significant illness due to COVID-19, as symptoms could be missed by the text search (or unrecorded by physicians) or individuals may be hospitalized due to secondary consequences of COVID-19 such as cardiovascular complications.

We have also revised the text to clarify that our endpoint of admissions with ≥ 1 acute respiratory infection-related diagnosis code likely provides the most specific indication of admissions associated with prominent respiratory complaints, but (similar to the circumstances mentioned above) may lack sensitivity since many severe COVID-19 complications may not be captured (lines 434-436).

To provide greater insight, we have cited data from several sources to address this point, noting that available US studies estimate 20-23% of all Omicron-associated admissions may be incidental, and that SARS-CoV-2 is expected to increase individuals' risk of hospital admission by a factor of 9.6-20.0 (or 2.8-5.9 when accounting for reduced risk of severe illness associated with the Omicron variant, lines 300-308), over the complete course of infection. We have added details on this calculation to Table S8 and its caption. Ultimately, these data support the interpretation that a solid majority of observed hospital admissions within the study cohort are due to COVID-19. As the proportion of hospital admissions associated with ARI is near equal for BA.2 and BA.4/BA.5 cases, differential misclassification of the endpoint between cases with either lineage is also unlikely.

4. Third alinea validation of SGTF. Instead of presenting the SGTF result among BA.2, BA.4 and BA.5 specimens, presenting the results as variants found within the SGTF and non-SGTF pools is more reasonable as this is the input for the model. For example BA.1 specimens may lead to an over estimation of BA.4/5 early on in the study. SO confirmation of PPV needs to be the other way around.

We have presented the validation as requested (probability that an infection is BA.2 given S gene detection, and probability that an infection is BA.4/BA.5 given SGTF) in the revised text (lines 81-84) and Table S2.

5. In addition the SGTF results needs to be validated by time period (if possible by week). As depending on the variant(s) circulating (non-)SGTF is a good proxy for BA.4/5 / BA.2. Did the authors use a cut-off value for Ct avlues? It is generally known that the S target is the first to become negative upon higher Ct values (lower viral load). This would lead to a false claim of SGTF, while just being a sensitivity issue for the three different targets. Therefor samples with a high Ct value need to be excluded.

We have included a monthly breakdown in Table S2 to show this assignment holds throughout the analysis period; numbers of isolates sequenced were sparse for weekly breakdowns. We have specified that positive specimens were those with at least two targets (among S, N, and Orf1a/b) having cycle threshold values <37 (those with cycle threshold values of ≥ 37 for ≥ 2 targets were excluded; lines 401-406). This is in contrast to several settings where values <40 are used to define positivity, although because specific sampling and assay conditions may affect the Ct count, head-to-head comparisons across settings are not necessarily meaningful. What is important to note is that throughout the study period, SGTF (per the above definition) had 95.5% predictive accuracy for identifying BA.4/BA.5 specimens (per whole genome sequencing), and S gene detection had 98.5% predictive accuracy for identifying BA.2 specimens. These results support use of SGTF to identify infecting lineages within the sample.

REVIEWER COMMENTS

Reviewer #1 (Remarks to the Author):

The revised manuscript has taken into account the remarks from the reviewer- and I have no further comments

Reviewer #3 (Remarks to the Author):

The authors have adequately addressed our remarks and suggestions.

One issue remains. The used cut-off for the TaqPath/SGTF data, [quote]:“We have specified that positive specimens were those with at least two targets (among S, N, and Orf1a/b) having cycle threshold values <37 (those with cycle threshold values of ≥ 37 for ≥ 2 targets were excluded; lines 401-406).”

Other papers using TaqPath SGTF data use more stringent thresholds ($Ct < 30$ or $Ct < 32$) to avoid sensitivity issues related to an early drop-out of S-target signal, despite the presence of the 69/70 residues. The authors only partially take away concerns about possible biases related to these thresholds. If among higher Ct values (lower viral loads) there is an unrightful classification of SGTF this might affect the analyses. A possible sensitivity analysis to confirm findings with a more stringent threshold would be an option, or sequence confirmation of such high Ct value samples, although the latter might be challenging regarding the sensitivity of sequence assays. It is generally known that sequence analyses is mostly successful for specimens with a Ct value below 32. It is currently unclear whether the authors have included such high Ct values in the sequence analyses as mentioned in the response to the reviewers [quote]: “What is important to note is that throughout the study period, SGTF (per the above definition) had 95.5% predictive accuracy for identifying BA.4/BA.5 specimens (per whole genome sequencing), and S gene detection had 98.5% predictive accuracy for identifying BA.2 specimens. These results support use of SGTF to identify infecting lineages within the sample”

Reviewer #1 (Remarks to the Author):

The revised manuscript has taken into account the remarks from the reviewer- and I have no further comments

We thank Reviewer 1 for their earlier helpful comments and for their assessment of the previous revision of the manuscript.

Reviewer #3 (Remarks to the Author):

The authors have adequately addressed our remarks and suggestions.

One issue remains. The used cut-off for the TaqPath/SGTF data, [quote]:“We have specified that positive specimens were those with at least two targets (among S, N, and Orf1a/b) having cycle threshold values <37 (those with cycle threshold values of ≥37 for ≥2 targets were excluded; lines 401-406).”

Other papers using TaqPath SGTF data use more stringent thresholds (Ct<30 or Ct<32) to avoid sensitivity issues related to an early drop-out of S-target signal, despite the presence of the 69/70 residues. The authors only partially take away concerns about possible biases related to these thresholds. If among higher Ct values (lower viral loads) there is an unrightful classification of SGTF this might affect the analyses. A possible sensitivity analysis to confirm findings with a more stringent threshold would be an option, or sequence confirmation of such high Ct value samples, although the latter might be challenging regarding the sensitivity of sequence assays. It is generally known that sequence analyses is mostly successful for specimens with a Ct value below 32. It is currently unclear whether the authors have included such high Ct values in the sequence analyses as mentioned in the response to the reviewers [quote]: “What is important to note is that throughout the study period, SGTF (per the above definition) had 95.5% predictive accuracy for identifying BA.4/BA.5 specimens (per whole genome sequencing), and S gene detection had 98.5% predictive accuracy for identifying BA.2 specimens. These results support use of SGTF to identify infecting lineages within the sample”

We thank the Reviewer for the opportunity to clarify this important point. No Ct cutoff is applied for which specimens are submitted for sequencing, as the diagnostic lab (unfortunately) does not retain quantitative Ct values after specimen processing. However, we have added data (Table S3, and lines 85-88) showing that the likelihood of sequencing failure was identical for SGTF and non-SGTF isolates submitted for sequencing, at 42.7% (998/2335) and 42.8% (1211/2830), respectively. This pattern holds during each month of the study period, as well as throughout the period overall. This observation in contrast to the situation the Reviewer is describing, where low viral RNA quantity would lead to sequencing failure in a higher share of SGTF specimens than non-SGTF specimens. Absent any such signal, we remain confident in the validity of SGTF as a basis for lineage determination in this study.

For clarity, we have also revised the manuscript to indicate that the data referenced in the quoted sentence are among specimens with a known sequence (line 83).